# Photoluminescence upconversion in monolayer WSe$_2$ activated by plasmonic cavities through resonant excitation of dark excitons

Niclas S. Mueller [1,9] ✉, Rakesh Arul [1], Gyeongwon Kang [1,2], Ashley P. Saunders[3], Amalya C. Johnson [4], Ana Sánchez-Iglesias [5,6], Shu Hu [1], Lukas A. Jakob[1], Jonathan Bar-David [1], Bart de Nijs [1], Luis M. Liz-Marzán [5,7,8], Fang Liu [3] & Jeremy J. Baumberg [1] ✉

Anti-Stokes photoluminescence (PL) is light emission at a higher photon energy than the excitation, with applications in optical cooling, bioimaging, lasing, and quantum optics. Here, we show how plasmonic nano-cavities activate anti-Stokes PL in WSe$_2$ monolayers through resonant excitation of a dark exciton at room temperature. The optical near-fields of the plasmonic cavities excite the out-of-plane transition dipole of the dark exciton, leading to light emission from the bright exciton at higher energy. Through statistical measurements on hundreds of plasmonic cavities, we show that coupling to the dark exciton leads to a near hundred-fold enhancement of the upconverted PL intensity. This is further corroborated by experiments in which the laser excitation wavelength is tuned across the dark exciton. We show that a precise nanoparticle geometry is key for a consistent enhancement, with decahedral nanoparticle shapes providing an efficient PL upconversion. Finally, we demonstrate a selective and reversible switching of the upconverted PL via electrochemical gating. Our work introduces the dark exciton as an excitation channel for anti-Stokes PL in WSe$_2$ and paves the way for large-area substrates providing nanoscale optical cooling, anti-Stokes lasing, and radiative engineering of excitons.

Anti-Stokes photoluminescence (PL) is a process in which light is emitted at a higher energy than the excitation laser by extracting energy from the material. The upconversion occurs through a variety of mechanisms including the absorption of phonons[1,2], Auger processes[3], or multi-photon absorption[4,5]. This leads to industrially-relevant applications in optical refrigeration[6], bioimaging[7], lasing[8], quantum information[9], and the detection of infrared light[10,11]. Excitons in two-dimensional semiconductors are a promising platform for

[1]NanoPhotonics Centre, Cavendish Laboratory, Department of Physics, University of Cambridge, Cambridge CB3 0HE, UK. [2]Department of Chemistry, Kangwon National University, Chuncheon 24341, South Korea. [3]Department of Chemistry, Stanford University, Stanford, CA 94305, USA. [4]Department of Materials Science and Engineering, Stanford University, Stanford, CA 94305, USA. [5]CIC biomaGUNE, Basque Research and Technology Alliance (BRTA), Paseo de Miramón 194, Donostia-San Sebastián 20014, Spain. [6]Centro de Física de Materiales, CSIC-UPV/EHU, Manuel Lardizabal Ibilbidea 5, Donostia-San Sebastián 20018, Spain. [7]Ikerbasque, Basque Foundation for Science, Bilbao 48009, Spain. [8]Centro de Investigación Biomédica en Red, Bioingeniería, Biomateriales y Nanomedicina (CIBER-BBN), Donostia-San Sebastián 20014, Spain. [9]Present address: Fritz Haber Institute of the Max Planck Society, 14195 Berlin, Germany. ✉e-mail: niclasmueller@gmx.de; jjb12@cam.ac.uk

anti-Stokes PL[2,5,12–14]. The reduced dielectric screening and enhanced Coulomb attraction make optical transition dipoles and exciton-phonon coupling an order of magnitude larger than in conventional bulk semiconductors or quantum wells[15–17]. Anti-Stokes PL in transition metal dichalcogenides (TMDs) is thus mostly explained by phonon-assisted processes and the interplay of different excitons[2,18,19].

Efficient anti-Stokes PL requires a condition in which both the excitation laser and the emission are resonant with a material excitation. Initial experiments on anti-Stokes PL in monolayer WSe₂ were explained by doubly resonant processes involving charged and neutral excitons[2,18], as well as A- and B-excitons[5]. Another possible excitation channel arises from the spin-orbit splitting of the conduction band in TMDs, which leads to the formation of bright and dark excitons with an energy splitting of several tens of meV (Fig. 1a)[19]. The dark excitons are either momentum-forbidden (intervalley) or spin-forbidden (intravalley) for excitation at normal incidence[15,16]. In WSe₂ and WS₂ they have lower energies than the bright excitons, which leads to quenching of the bright exciton emission at low temperature due to fast non-radiative relaxation to the dark exciton[20,21]. The energetic ordering makes dark excitons on the other hand a potential excitation channel for anti-Stokes PL. Excitation with far-field radiation is however inefficient because the dark intravalley exciton has an out-of-plane transition dipole[22,23]. Promising routes to activate the dark exciton are through strong magnetic fields or through metallic nanostructures that sustain surface plasmons[24–26]. The optical near-fields of plasmonic antennas enable direct coupling to the out-of-plane transition dipole of the dark exciton and enhance the normal Stokes PL emission, which was demonstrated for metallic tips as well as nanoparticle-on-mirror cavities[25,27,28].

Here, we show that plasmonic cavities activate anti-Stokes PL in WSe₂ through resonant excitation of the dark intravalley exciton. We employ nanoparticle-on-mirror cavities to couple to the out-of-plane transition dipole of the dark exciton as an excitation channel, and enhance the outcoupling through the bright exciton as an emission channel. Through a statistical analysis of hundreds of plasmonic cavities, we demonstrate that coupling to the dark exciton is key to enhancing the anti-Stokes PL emission by two orders of magnitude. This is further corroborated by experiments in which we tune the excitation wavelength. We show that decahedral nanoparticles with asymmetric shapes and precise Au(111) facets are one of the advantageous geometries to activate the dark exciton and achieve consistent enhancement of the anti-Stokes PL. Finally, we demonstrate a selective switching of the upconverted PL in an electrochemical cell.

## Results
### Plasmonic nanoparticle-on-mirror cavities
We prepare mm-sized monolayers of WSe₂ by gold tape exfoliation of bulk van der Waals crystals[29,30]. The WSe₂ monolayers are first exfoliated on a 285 nm SiO₂/Si substrate and then transferred back onto a template-stripped Au surface with polymer to prevent direct binding with the Au which quenches the PL emission (SI Section S1). Au nanoparticles are then deposited on top of the WSe₂ to form nanoparticle-on-mirror (NPoM) plasmonic cavities (Fig. 1b)[31,32]. The NPoM cavities confine light to the nm gap between a facet of the nanoparticle and the Au mirror enclosing the WSe₂ monolayer, which leads to a near 10⁴-fold enhancement of the local light intensity (Fig. 1b, right, and SI Section S7). The plasmonic near fields have both in-plane and out-of-plane components which allow coupling to the bright

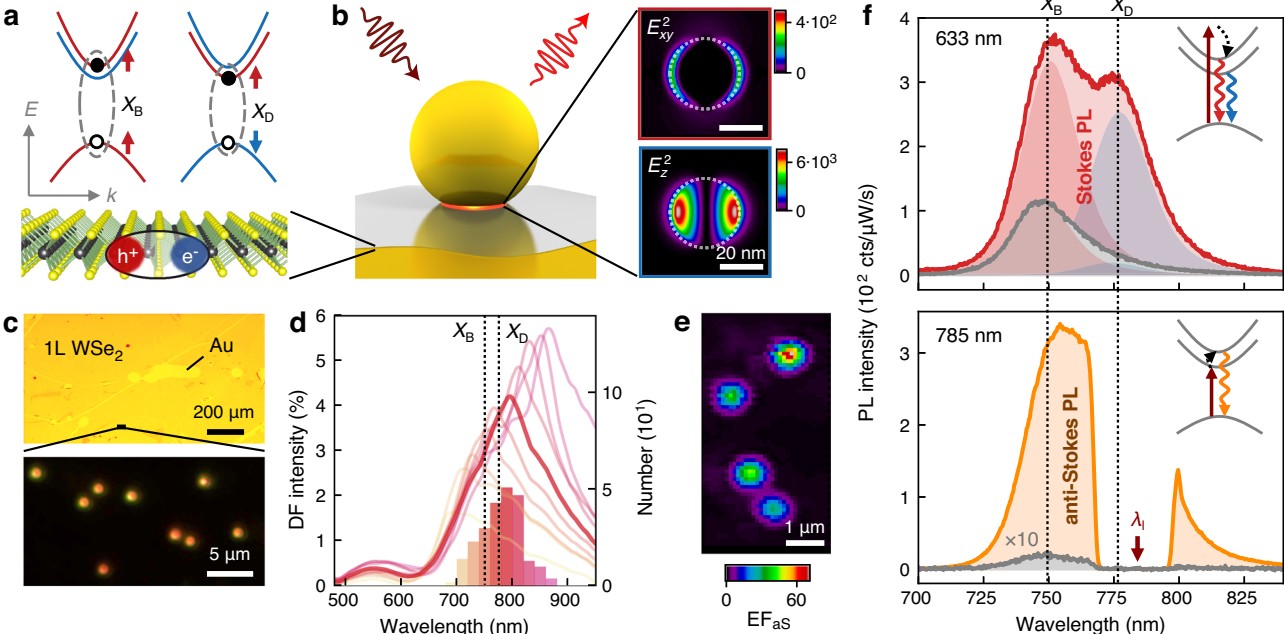

**Fig. 1 | Activation of dark exciton and anti-Stokes PL in WSe₂ by plasmonic cavities. a** Band-structure of spin-allowed bright ($X_B$) and spin-forbidden dark ($X_D$) A excitons in WSe₂. **b** Nanoparticle-on-mirror (NPoM) plasmonic cavity with WSe₂ defining a gap between a bottom facet of the nanoparticle and the Au mirror. Simulated field intensity enhancement in the WSe₂ monolayer is shown on the right, with in-plane $E_{xy}^2$ and out-of-plane $E_z^2$ enhancement of the field amplitude $E$. Dotted lines show edges of the nanoparticle bottom facet (see Fig. S8 for geometry). **c** Optical bright-field micrograph of large-area monolayer WSe₂ on Au (top), and dark-field micrograph of NPoM cavities on WSe₂ (bottom). **d** Histogram of dark field scattering spectra of plasmonic NPoM cavities from measurements on 252

cavities. Left axis refers to the average spectra for each bin and the right axis to the histogram. **e** Map of anti-Stokes PL by scanning $\lambda_l$ = 785 nm excitation laser across four cavities with precise decahedral nanoparticle shapes (see Fig. 4 below). **f** Stokes PL spectrum with a laser excitation wavelength $\lambda_l$ = 633 nm (top, red) and anti-Stokes PL spectrum with $\lambda_l$ = 785 nm (bottom, orange). Reference spectra recorded beside plasmonic cavities are shown in gray. Peak components from fits of bright (red) and dark (blue) excitons are shown for Stokes PL. Insets show energy diagrams of the Stokes (red, blue) and anti-Stokes PL (orange) emission, with excitation laser (dark red) and internal energy conversion (black dotted). Dotted lines show energies of bright and dark excitons.

exciton ($X_B$) with an in-plane transition dipole as well as the dark exciton ($X_D$) with an out-of-plane transition dipole (Fig. 1a, b)[28].

Our simple approach to sample preparation allows the simultaneous fabrication of thousands of NPoM cavities encapsulating WSe$_2$ monolayers, located from their dark-field scattering pattern in an optical microscope (Fig. 1c). Using particle recognition algorithms and a fully automated setup we measure the optical spectra of hundreds of plasmonic cavities. The dark-field scattering spectra show a broad plasmonic mode with a central wavelength between 740–820 nm for most of the NPoM cavities, which overlaps with the expected emission wavelengths of the dark and bright excitons of WSe$_2$ (Fig. 1d). Variations of the confined plasmon wavelength partly originate from different nanoparticle shapes and geometries of the bottom facet (see below)[33].

### Plasmonic activation of dark exciton and PL upconversion
The PL emission of WSe$_2$ is enhanced when centering the laser measurement spot on a NPoM cavity (Fig. 1e, f). We choose two excitation laser wavelengths $\lambda_l$ to yield Stokes PL ($\lambda_l = 633$ nm) and anti-Stokes PL ($\lambda_l = 785$ nm). The Stokes PL is enhanced by a factor of 4.3 compared to the signal recorded beside the plasmonic cavity (Fig. 1f, top). Most interestingly, a new emission peak appears at 777 nm that is shifted by ~50 meV from the bright exciton $X_B$ at 750 nm. We attribute this additional peak to PL emission from the dark intravalley exciton $X_D$, as also reported in recent work on NPoM cavities (Fig. 1f, top, inset)[28]. The dark exciton has an out-of-plane transition dipole that couples selectively to the optical field components in the plasmonic cavity that point normally to the WSe$_2$ monolayer[25,27,28]. As the plasmonic near-fields in the nm gap of NPoM cavities are primarily polarized out-of-plane (Fig. 1b, right), the NPoM geometry couples more efficiently to the dark exciton than the bright exciton[28]. Coupling to momentum-indirect dark excitons is less efficient because of their in-plane transition dipole and momentum matching with plasmonic nanostructures would require field confinement on the sub-nm scale[34]. Such momentum-indirect excitons can also mediate upconversion but are not selectively activated by the plasmonic cavities here[19]. We also exclude a dominant contribution from charged excitons here as we see no signature of photodoping in the excitation power dependence and all experiments were conducted at room temperature (SI Section S3)[35,36]. We observe an average dark-bright splitting $\Delta E_{DB} = 55 \pm 8$ meV that is consistent with previous measurements at room temperature with $\Delta E_{DB} \approx 40$–60 meV (Fig. S6)[25,27,28]. Variations can occur because of a different dielectric screening and image dipole interaction of dark and bright excitons at metal interfaces[28,37]. Weak emission from the dark intravalley exciton is already visible in the reference spectrum recorded beside the NPoM cavity because we collect the PL emission with a high numerical aperture objective (Fig. 1f, top, gray)[22]. Comparison to the reference spectrum allows us to estimate an enhancement factor EF$_D$ = 12 for the dark exciton, while the bright exciton is only enhanced by EF$_B$ = 3 (see peak components in Fig. 1f top). As exciton-phonon coupling also contributes to the asymmetric lineshape of the reference spectrum, the enhancement of the dark exciton is probably underestimated[38].

The enhancement of the PL emission becomes an order of magnitude larger when moving to an excitation wavelength of $\lambda_l = 785$ nm that leads to anti-Stokes PL emission, as this is near-resonant with the dark exciton (Fig. 1f, bottom). The enhancement EF$_{aS}$ = 170 is so large that the anti-Stokes PL for $\lambda_l = 785$ nm becomes comparable to the Stokes emission for $\lambda_l = 633$ nm. From comparison to Stokes PL emission on glass, we estimate a quantum yield of the enhanced upconverted PL on the order of 0.1% (SI Section S5)[39]. The PL enhancement is tightly localized to the position of the plasmonic cavities (Fig. 1e). From line scans of the excitation laser spot across single NPoM cavities we extract a spatial FWHM-460 $\pm$ 30 nm of EF$_{aS}$, close to the diffraction-limited laser spot size $\lambda_l/2NA = 440$ nm (Fig. S2). This is different from previous work,

where the PL signal was only slightly reduced several micrometers away from the plasmonic cavities[13]. When considering the size mismatch of the laser excitation spot with the area of the nanoparticle bottom facet where enhancement occurs, we estimate local enhancement factors of EF$_{aS, loc}$ (785 nm) $\approx 9 \cdot 10^4$, EF$_{D, loc}$ (633 nm) $\approx 4 \cdot 10^3$, and EF$_{B, loc}$ (633 nm) $\approx 9 \cdot 10^2$ for the spectra in Fig. 1f (SI Section S4). The difference in enhancement at 785 nm and 633 nm excitation cannot be explained by the wavelength dependence of the plasmonic near-field intensity when only considering the bright exciton (SI Section S7). This hints at the dark exciton as an additional excitation channel for anti-Stokes PL (Fig. 1f, bottom, inset).

### Role of dark exciton for PL upconversion
To test this hypothesis, we characterize the PL enhancement of 252 NPoM cavities (Fig. 2). For each plasmonic cavity we correlate the enhancement of the anti-Stokes PL with both dark and bright excitons in the Stokes PL emission (Fig. 2a). Almost all NPoM cavities enhance the PL emission, with the dark exciton enhancement $\langle EF_D \rangle = 7$ on average larger than that of the bright exciton $\langle EF_B \rangle = 3$. The enhancement factors, however, vary by an order of magnitude, and not all NPoM cavities activate the dark exciton. The variation in anti-Stokes PL enhancement is even larger, ranging from EF$_{aS} = 1$ to >200, with an average $\langle EF_{aS} \rangle = 20$. The average enhancement agrees well with finite-difference time-domain (FDTD) simulations which predict EF$_{aS} = 21$, where the enhancement mostly originates from the increased incoupling through the dark exciton (SI Section S7). To better understand the large spread in enhancements, we compare the average PL spectra of NPoMs that activate the dark exciton (EF$_D$/EF$_B \geq 2$, Fig. 2b left) with NPoMs that do not activate the dark exciton (EF$_D$/EF$_B$<2, Fig. 2b right). Indeed, when the dark exciton is activated the average enhancement of the anti-Stokes PL is larger $\langle EF_{aS} \rangle = 65$ than for NPoM cavities that do not couple to the dark exciton $\langle EF_{aS} \rangle = 21$. This hints at a mechanism in which (i) the 785 nm laser excites the dark exciton, (ii) energy is extracted from the material, e.g., through phonon-assisted processes, and (iii) anti-Stokes emission finally occurs via the bright exciton (Fig. 2c). Plasmon-mediated coupling to the dark exciton is thus a key mechanism to enhance the anti-Stokes PL. The activation of the dark exciton EF$_D$/EF$_B$, however, varies by one order of magnitude and only a subset (30%) of the NPoM cavities enhances the dark exciton more than ×2, which explain the large spread of EF$_{aS}$ (Fig. 2d and S7). We discuss how to improve this below.

### Excitation wavelength dependence of PL upconversion
To better understand the excitation mechanism of anti-Stokes PL, we sweep the laser excitation wavelength from $\lambda_l = 730$–805 nm across the resonances of the dark and bright excitons and measure excitation-emission profiles (Fig. 3). The emission evolves from Stokes PL at $\lambda_l = 730$ nm to anti-Stokes PL at $\lambda_l = 805$ nm (Fig. 3a). Without plasmonic cavities, the Stokes emission is largest when exciting blue-detuned from the bright exciton, while anti-Stokes emission is largest when resonantly exciting the bright exciton (Fig. 3b, top). This is expected, as the bright exciton is the only excitation channel. The excitation-emission profile changes in the presence of a plasmonic NPoM cavity (Fig. 3b, bottom). An additional resonance appears at excitation wavelengths red-detuned from the bright exciton, and the anti-Stokes emission is largest for $\lambda_l$ close to the dark exciton. This is also consistently seen in the excitation-emission profiles of other NPoM cavities (Fig. 3c) and shows that the dark exciton serves as an excitation channel for anti-Stokes PL. The excitation wavelength dependence also shows that anti-Stokes electronic Raman scattering is not a dominant emission channel here, as the emission peak does not shift with $\lambda_l$ (Fig. 3a). A small redshift (15 meV) is visible with increasing $\lambda_l$, which we attribute to the upconversion mechanism discussed below.

The anti-Stokes PL is most efficient for excitation wavelengths between the dark and bright excitons, as the upconversion process

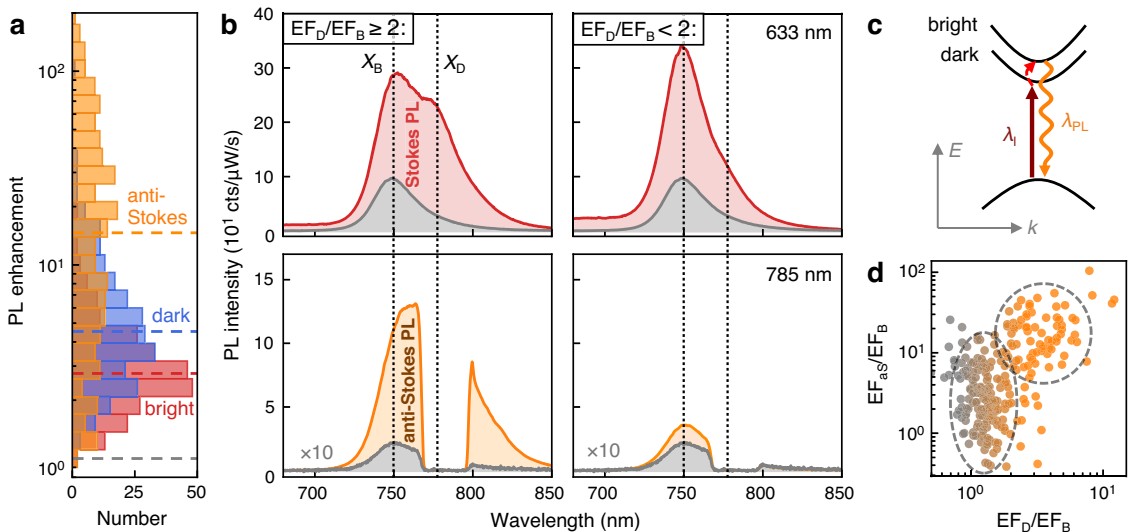

**Fig. 2 | Correlation between dark exciton activation and anti-Stokes PL enhancement. a** Histogram of PL enhancement factors from measurements on 252 NPoM cavities. Enhancement of anti-Stokes PL (EF$_{aS}$, orange) from $\lambda_l = 785$ nm pumping, and enhancement of dark (EF$_D$, blue) and bright (EF$_B$, red) excitons from $\lambda_l = 633$ nm pumping. Dashed lines show median enhancement factors. **b** Average Stokes (top, red) and anti-Stokes (bottom, orange) PL spectra for NPoMs that activate the dark exciton ($\langle$EF$_{aS}\rangle = 65$, left) vs. NPoMs that do not activate the dark exciton ($\langle$EF$_{aS}\rangle = 21$, right). Reference spectra in gray. **c** Anti-Stokes PL process via resonant excitation of dark exciton; red dashed arrow shows coupling to phonons. **d** Relative strength of anti-Stokes PL enhancement EF$_{aS}$ vs. activation of dark exciton EF$_D$, each normalized to bright exciton EF$_B$. Dashed lines show the clustering of data points for NPoMs that couple (orange) or not (gray) to the dark exciton.

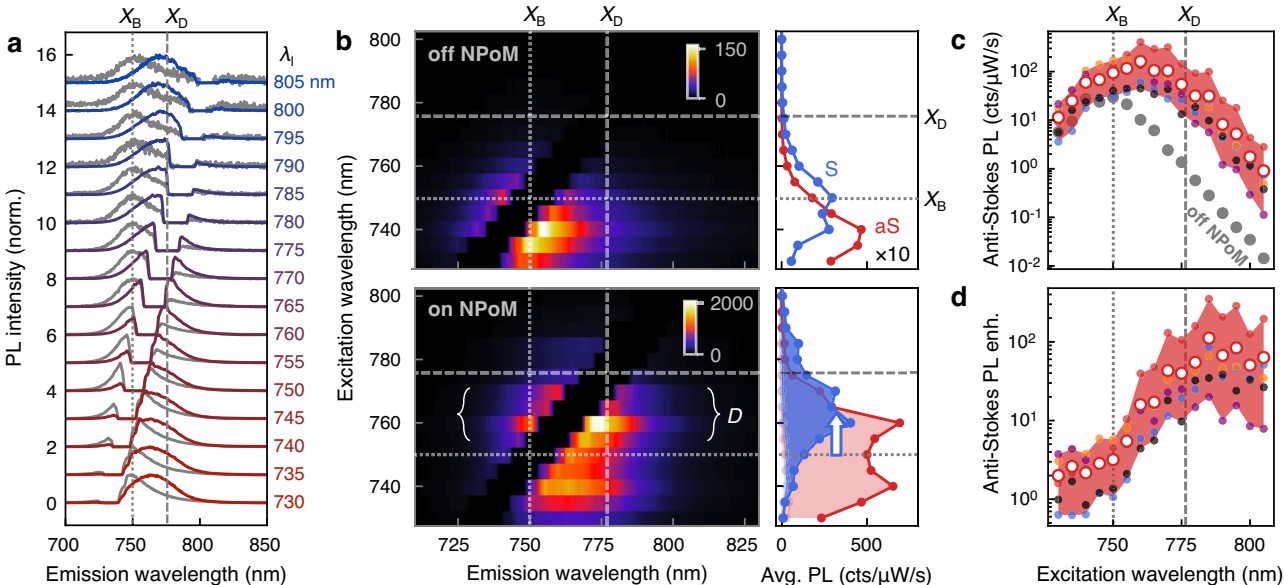

**Fig. 3 | Excitation wavelength dependence. a** Normalized PL spectra enhanced by NPoM cavity (colored) and beside cavity (gray) while tuning excitation wavelength $\lambda_l$ from 730 nm (bottom) to 805 nm (top). **b** Excitation emission map of PL spectra measured beside cavity (top) and enhanced by NPoM cavity (bottom). Color bars are in counts/µW/s. Panels on the right show the average signal of anti-Stokes (aS, blue) and Stokes PL (S, red). Label '*D*' refers to a new resonance activated by dark exciton. Colored areas show differences to reference. **c** Anti-Stokes PL signal and **d** enhancement by five NPoM cavities (small colored dots), their average (red circles), and range (red area), and reference measurement beside cavities (gray). Dashed lines show wavelengths of dark and bright excitons.

requires energy to be extracted from the material, which becomes less efficient with increasing energy difference of excitation and emission (Fig. 3b, c). The upconversion can occur either through the coupling to thermally populated phonons, or electron-electron scattering[2,19,20]. Nonlinear excitation, e.g., through two-photon absorption[40], can be excluded here, as we observe a near-linear power dependence of the anti-Stokes PL at these ~10 µW CW excitation powers (SI Section S3). Electron-electron scattering can be potentially enhanced by plasmons through an ultrafast charge transfer of hot electrons into WSe$_2$ which

leads to an increase of the electronic temperature[41]. This is however unlikely here as WSe$_2$ and Au are not in direct contact and we do not observe signatures of photodoping in the excitation power dependence (Fig. S3c, f)[35]. The energy shift between $\lambda_l$ for largest anti-Stokes PL and the bright exciton (20–40 meV) matches the energies of optical phonons in WSe$_2$ (Fig. 3c). This hints to a phonon-mediated upconversion, e.g., through the chiral E″ phonon that can couple the dark and bright intra-valley excitons, as well as multi-phonon processes for larger energy shifts[19,36,42–45]. Such phonon-assisted upconversion

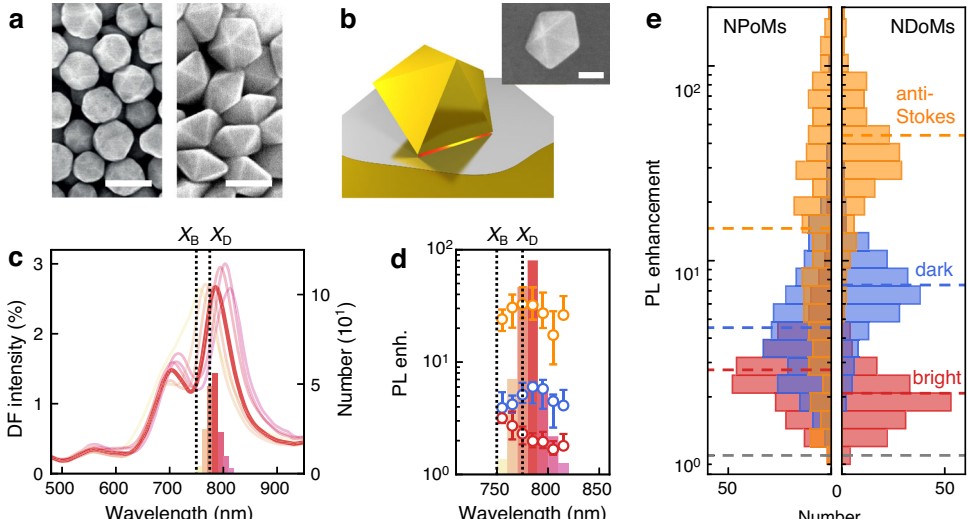

**Fig. 4 | Consistent enhancement with nano-decahedral cavities. a** SEM image of nanoparticles used for NPoM cavities above (left, adapted from ref. 46, ACS under CC-BY license) vs. decahedral nanoparticles with precise shape and size (right). Scale bars are 100 nm. **b** Sketch of nanodecahedra-on-mirror (NDoM) plasmonic cavity. The inset shows an SEM image of NDoM (scale bar 50 nm). **c** Histogram of dark field (DF) scattering spectra of 173 NDoM cavities, where the left axis refers to average spectra for each bin and right axis to histogram. **d** Median of $EF_{aS}$ (orange), $EF_D$ (blue), and $EF_B$ (red) vs. DF peaks in **c**. Error bars are from the first to the third quartile. **e** Histograms of enhancement factors for NPoMs (left) vs. NDoMs (right). Dashed lines show median EFs.

requires a thermal population of phonons and is therefore most efficient at room temperature or above, as demonstrated in previous experiments[2,13,18,36,45]. The overall excitation wavelength dependence is determined by an interplay of plasmon-enhanced excitation of the dark exciton and upconversion through phonons and other mechanisms. To extract the role of the plasmonic cavities we determine the anti-Stokes PL enhancement for each excitation wavelength (Fig. 3d). In contrast to the overall PL emission, the enhancement factor peaks at excitation wavelengths slightly red detuned from the dark exciton at $\lambda_l \approx 785$ nm, as used here in all other measurements. The aS-PL is enhanced by $\geq 100$ for wavelengths beyond the dark exciton that match the plasmon resonances of the NPoM cavities.

## Consistent enhancement with nano-decahedral cavities

Next, we address the large variation in anti-Stokes PL enhancement factors (see Fig. 2a, d). A similar spread in enhancement was also observed in surface-enhanced Raman scattering from NPoM cavities and was attributed to variations in nanoparticle shape and size (Fig. 4a, left)[46]. The plasmonic resonances of NPoM cavities mainly depend on the geometry of the nanoparticle's bottom facet[33], which varies considerably for the nanoparticles used here. We therefore employ decahedral nanoparticles with precise geometry and triangular Au(111) facets (Fig. 4a, right)[47]. These were recently demonstrated to yield more consistent SERS enhancement and plasmonic modes when assembled as nanodecahedra-on-mirror (NDoM) cavities (Fig. 4b)[48]. Indeed, we observe a three-fold narrower distribution of plasmon resonance wavelengths than for NPoM cavities (compare Figs. 4c and 1d). The dominant NDoM plasmonic mode centered around 780 nm for this size is chosen to spectrally overlap with the dark exciton of WSe₂. The plasmonic resonances are spectrally narrower than for NPoM cavities and arise from higher-order plasmonic modes that are located under the bottom facet of the nanoparticles (Fig. S11)[48].

The precise geometry of the NDoM cavities leads to a much more consistent enhancement of the PL emission of WSe₂ (Fig. 4e). Compared to NPoMs the distribution of enhancement factors is two-fold narrower for Stokes PL and ten-fold for anti-Stokes PL. The NDoM cavities consistently activate the dark exciton with $\langle EF_D \rangle = 7.5$ and $\langle EF_B \rangle = 2$, while 90% of the cavities provide $EF_D/EF_B > 2$. This leads to an enhancement of the anti-Stokes PL $\langle EF_{aS} \rangle = 44$ that is on average larger

than for NPoM cavities which give $\langle EF_{aS} \rangle = 20$, while the enhancement is similar to that of NPoMs activating the dark exciton (Fig. S7b). Given that NPoM cavities consist of a broad range of nanoparticle geometries, the large enhancement by NDoM cavities shows that this is one of the advantageous geometries to enhance the dark exciton and anti-Stokes PL. The measured enhancement is in good agreement with FDTD simulations for the Stokes PL but underestimated for anti-Stokes PL (SI section S7-3). Possible reasons are cancellations of different emission channels in our simulations, a different dark-to-bright conversion on and beside the nanoparticle, and effects of exciton diffusion that are not included in our simulations. We attribute the remaining spread in enhancement factors to intrinsic spatial variations of the WSe₂ PL emission that occur across the substrate (Fig. S6), and the distribution of plasmon resonance wavelengths. Indeed, when sorting the enhancement factors by the distribution of plasmon resonance wavelengths, we find that $EF_D$ and $EF_{aS}$ are largest when the plasmon resonance wavelength matches the dark exciton, which is the case here for almost all NDoMs (Fig. 4d, blue and orange). The bright exciton is on the other hand detuned from the plasmon resonances leading to a smaller $EF_B$ than for NPoMs (Fig. 4d, e, red). Since the most efficient anti-Stokes PL requires bright and dark exciton to be both resonant with plasmonic modes, this gives room for further optimization.

## Active electrochemical switching of PL upconversion

Finally, we show how to actively modulate the upconverted PL emission with electrochemical gating. We use a custom-built spectro-electrochemical cell that allows us to record PL spectra of individual plasmonic cavities in a liquid electrolyte (Fig. 5a, top)[49,50]. A potential bias between a Pt electrode and the Au mirror leads to the formation of an electric double layer, which results in a static electric field across the WSe₂ monolayer and charge doping, similar to field effect transistors and ionic liquids (Fig. 5a, bottom)[51,52]. We apply small voltages to avoid shifting the Fermi level above the conduction band edge of WSe₂, which would lead to charge injection and the formation of charged excitons[51]. When sweeping the potential between +0.2 V and −0.4 V we observe a ≈50% modulation of the anti-Stokes PL, enhanced by the NDoM cavity (Fig. 5b). This also works reversibly on the sec timescale when step-wise switching the voltage (Fig. 5c, e). The modulation is almost as large as in the much weaker spectra recorded beside an

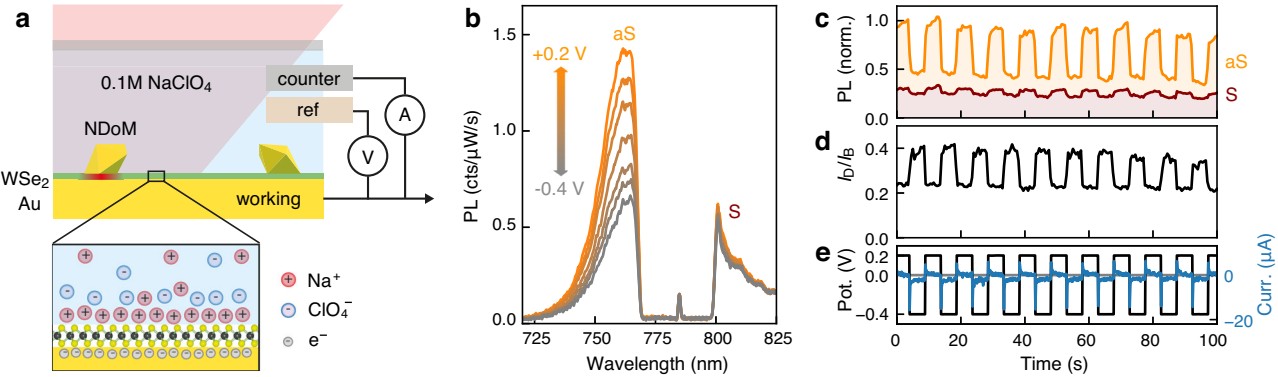

**Fig. 5 | Selective electrochemical switching of PL upconversion. a** Sketch of spectro-electrochemical cell (top) and electric double layer at the Au-WSe$_2$-electrolyte interface (bottom). **b** PL spectra measured with $\lambda_l = 785$ nm on NDoM cavity while continuously sweeping the potential from +0.2 V to −0.4 V and back (spectra averaged for each voltage interval). Modulation on the anti-Stokes side (aS) is much larger than on the Stokes side (S). **c** Modulation of PL intensity on aS and S sides with $\lambda_l = 785$ nm, and **d** of the intensity ratio of dark and bright excitons with $\lambda_l = 633$ nm, while **e** step-wise switching the potential (black) and measured electrical current (blue). Data in **c**, **d** are recorded on the same NDoM cavity.

NDoM cavity (Fig. S12), which shows that ions penetrate into the gap of the plasmonic cavity where the upconversion occurs[49,50]. We attribute the decrease in PL at negative potentials to charge screening and non-radiative Auger recombination of the excitons in WSe$_2$[51,53]. Interestingly, the modulation is much larger on the anti-Stokes side ≈50% than on the Stokes side ≈15% (Fig. 5b, c). In the enhanced PL spectra recorded with $\lambda_l = 633$ nm, we observe a stronger modulation of the bright exciton intensity than of the dark exciton intensity (Fig. S13), which is consistent with previous measurements using field-effect transistor geometries[26,43]. The dark/bright intensity ratio is modulated by ≈40%, with a relative increase of the dark exciton intensity at negative potentials (Fig. 5d). This compensates for the quenching on the Stokes side of $\lambda_l = 785$ nm emission spectra and enables a selective modulation of the upconverted PL.

## Discussion

In conclusion, we showed that plasmonic coupling to the dark exciton in WSe$_2$ activates an efficient excitation channel for anti-Stokes PL. This leads to a near hundred-fold enhancement of PL upconversion that occurs within spatially confined fields of plasmonic cavities. Compared to previous work, our large-area samples allow us to analyze the optical spectra of hundreds of plasmonic cavities and correlate the enhancement factors of different emission channels. We showed that a precise nanoparticle shape is important for consistent enhancement and that decahedral nanoparticles are one of the advantageous geometries to achieve large enhancement factors. The upconverted PL can be selectively modulated through electrochemical gating, which paves the way for active devices, superresolution imaging, and nanoscale thermal management. Our approach may be extended to localized excitons at excitation wavelengths below the dark exciton, that are activated through local strain or defects, and to other 2D luminescent materials[26,54–56]. The simplicity and scalability of our sample fabrication pave the way for large-area substrates for PL upconversion, which may find applications in nanoscale optical refrigeration of 2D materials, anti-Stokes lasing, and radiative engineering of excitons in the future.

## Methods

### Sample preparation

**Exfoliation of WSe$_2$.** WSe$_2$ monolayers are exfoliated with the gold tape exfoliation method from bulk WSe$_2$ single crystals (HQ Graphene)[29]. In brief, a 100 nm Au layer is evaporated on a Si wafer with an ebeam evaporator. The Au surface is spin-coated with a layer of polyvinylpyrrolidone (PVP) as a protective layer against further contaminations from processing. The PVP/Au stack is stripped from the Si surface with a thermal release tape (Nitto). The template-stripped Au

surface is pressed onto the cleaved WSe$_2$ single crystal, exfoliating a WSe$_2$ monolayer on the Au surface. The PVP/Au/WSe$_2$ stack is released onto a 285 nm SiO$_2$/Si substrate upon heating. The PVP protection layer is dissolved with water, and the Au is etched with KI/I$_2$ solution, obtaining a clean WSe$_2$ monolayer on the SiO$_2$/Si substrate. The WSe$_2$ monolayers are transferred with cellulose acetate butyrate (CAB) polymer from SiO$_2$/Si to a new template-stripped Au substrate using a wedging transfer technique[57]. Finally, the CAB polymer is removed with ethyl estate.

**Preparation of NPoM cavities.** For the preparation of NPoM cavities, 80 nm Au nanoparticles (BBI Solutions, mean diameter 77–85 nm) were mixed with 0.1 M NaNO$_3$ (10:1), drop cast on a selected area of the sample, and washed off with DI water after 10 s. Nanoparticles that remain on the substrate form NPoM cavities and were identified with a dark field microscope and spectrometer. Following similar steps as for the preparation of NPoMs, the decahedral nanoparticles were drop cast onto the WSe$_2$ substrate to form plasmonic NDoM cavities[48].

**Synthesis of gold decahedra.** Decahedral Au nanoparticles were synthesized using seed-mediated growth[47,48]. In brief, penta-twinned gold seeds were used to grow decahedral nanoparticles in a solution containing benzyldimethylhexadecyl-ammonium chloride (BDAC, 50 mL, 100 mM), gold (III) chloride trihydrate (HAuCl$_4$, 0.5 mL, 50 mM, ≥99.9%) and ascorbic acid (AA, 0.375 mL, 100 mM, ≥99%) to an edge length of 41 nm. Gold decahedra with an edge length of 70 nm were then synthesized by adding 41 nm gold decahedra seeds (0.5 mL, 5 mM) to a growth solution containing BDAC (50 mL, 15 mM), HAuCl$_4$ (0.25 mL, 50 mM) and AA (0.19 mL, 100 mM). The gold decahedra solution was centrifuged at the different growth steps to remove excess reactants and dispersed in aqueous hexadecyl-trimethylammonium bromide (CTAB) solution (10 mM). Finally, the decahedra were functionalized with citrate through subsequent centrifugation and redispersion in poly sodium 4-styrenesulfonate (Na-PSS, 50 mL, 0.15 % wt, $M_w$ ~ 70000 g/mol), and finally sodium citrate (50 mL, 5 mM, ≥99%).

**Photoluminescence and dark field spectroscopy**

Optical spectra were recorded with custom-built setups that consist of a dark field microscope with a motorized stage and spectrometers. The setups are fully automated to record the optical spectra of hundreds of plasmonic cavities. We used particle recognition algorithms to locate NPoM cavities by their dark field scattering pattern and center them to the measurement spot. Darkfield (DF) scattering spectra were

recorded by illuminating the sample with incoherent white light (halogen lamp) through an Olympus MPLFLN 100x DF objective with 0.9 NA and recording the scattered light with a fiber-coupled QE Pro spectrometer (Ocean Optics). For each NPoM cavity, an automatic scan through the focal depth is performed, and the depth-dependent DF intensity is used to correct the spectrum from chromatic aberrations.

For PL spectroscopy we used either CW single-frequency diode lasers with $\lambda_l = 632.8$ nm (Integrated Optics, MatchBox) and $\lambda_l = 785$ nm (Thorlabs, LP785-SF100), or a tuneable cw Ti:Sapphire Laser (SolsTiS, M Squared Lasers). The lasers were focussed to a diffraction-limited laser spot through the 100x DF objective and the laser power on the sample was kept below 20 μW, unless stated otherwise. The typical CW power of 10 μW used here amounts to ≈300 mW/μm² peak intensity in the hotspot of the plasmonic cavity. This is far below the typical peak powers of pulsed lasers and we therefore avoid exciton saturation. The emitted PL was collected with the same objective and guided to a grating spectrometer (Andor Kymera or Shamrock, 150 l/mm grating) with a CCD camera (Andor Newton EMCCD). The spectrometer slit width and readout lines on the CCD were limited to detect PL from a single nanoparticle. For automated measurements comparing 633 nm and 785 nm excitation, the two lasers were combined with a dichroic mirror and subsequently switched on. Pairs of notch filters for the two laser wavelengths were subsequently moved into the detection path with automated sliders. For measurements with tuneable laser excitation, we used angle-tuneable laser clean-up filters and notch filters. All spectra were dark count subtracted with spectra recorded on the bare Au substrate. The datasets for statistical measurements on NPoM and NDoM cavities were pre-screened, and data were discarded from further analysis if the excitation laser was defocussed or shifted from the nanoparticle, or if the nanoparticle was beside $WSe_2$. Reference spectra were recorded 2 μm beside each plasmonic cavity and the average spectra from all positions were used to calculate enhancement factors. All measurements were conducted at room temperature.

### Scanning electron microscopy

For SEM characterization, Au nanoparticles were cleaned three times with ultrapure water by centrifugation (3000 rpm) to remove as much of the surfactant as possible. 3 μl concentrated Au nanoparticle solution was dropped on a silicon substrate which was then dried in a vacuum chamber. The SEM measurement was performed on a Hitachi S4800 with a 10 keV accelerating voltage.

### Finite-difference time-domain simulations

To simulate the electric field enhancement of NPoM cavities, we used the software package Lumerical FDTD Solutions from Ansys. We identified a geometry of an NPoM cavity that is representative of our experiments by reproducing the experimental scattering spectra (Fig. S8). The nanostructure was illuminated from the top with a Gaussian beam source with 0.9 NA and the electric field intensity in the NPoM gap was recorded with an electric field monitor. The electric fields were normalized by the total electric fields without the plasmonic nanostructure to calculate enhancement factors. All further details of the simulations are provided in the Supporting Information (SI section S7).

### Electrochemical gating

We used a custom-built 3-electrode spectro-electrochemical cell[49,50]. The cell was filled with a 0.1 M aqueous solution of $NaClO_4$ as an electrolyte. The Au mirror of the template-stripped Au sample was used as a working electrode. A Pt wire was used as the counter electrode and Ag/AgCl as the reference electrode. PL was excited and detected through a glass cover slip on top of the cell with a 0.9 NA microscope objective.

## Data availability

The figure source data generated in this study have been deposited in the Cambridge open data archive under the accession code https://doi.org/10.17863/CAM.101514.

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

## Acknowledgements

The authors acknowledge funding from the EPSRC (EP/L027151/1 and EP/R013012/1), and the EU (883703 PICOFORCE, 861950 POSEIDON). B.d.N. acknowledges support from the Winton Program for the Physics of Sustainability and from Royal Society University Research Fellowship URF\R1\211162. L.M.L.-M. acknowledges funding from the Spanish Ministerio de Ciencia e Innovacion, MCIN/AEI/10.13039/501100011033 (Grant PID2020-117779RB-100). N.S.M. acknowledges support from the German National Academy of Sciences Leopoldina. R.A. acknowledges support from the Rutherford Foundation of the Royal Society Te Apārangi of New Zealand, the Winton Program for the Physics of Sustainability, and Trinity College Cambridge. L.A.J. acknowledges support from the Cambridge Commonwealth, European & International Trust, and EPSRC award 2275079. J.B.D. acknowledges support from the Blavatnik fellowship. F.L. acknowledges support from the Terman Fellowship and startup funds from the Department of Chemistry at Stanford University. We thank Angela Demetriadou and Demelza Wright for the helpful discussions.

## Author contributions

N.S.M. and J.J.B. conceived the project. N.S.M., R.A., L.J., J.B.-D and B.d.N. designed the optical setups, N.S.M. and R.A. conducted the optical experiments, G.K. electrochemical gating, and S.H. electron microscopy. A.P.S., A.C.J. and F.L. prepared the $WSe_2$ samples. A.S.-I. and L.M.L.-M. synthesized the decahedral nanoparticles. N.S.M. analyzed the data, which were discussed with all coauthors. The manuscript was written with contributions from all authors.

## Competing interests

The authors declare no competing interests.
