## [Peer Review File · Nature Communications]

Reviewers' Comments:

Reviewer #1:

Remarks to the Author:

In this article Mueller et al. and co-workers demonstrated up-converted photoluminescence process in monolayer WSe₂ based on plasmonic nanocavities through resonant excitation of dark exciton. Author demonstrated that how a tightly confined plasmonic field excite the dark exciton that leads to light emission from bright exciton at higher energy. Furthermore, author showed that coupling to dark exciton leads to hundred -fold boost of up converted PL intensity. Overall, this could be an interesting finding and may helpful futuristic application in the field of material science and physics. But there are existing reports where such system (i.e. based on plasmonic nanocavity concept) were already addressed where authors demonstrated less than 1000-fold enhancement of up -converted PL due to Purcell effect (author already cited this article in ref 13. Note in this reference some authors are missing too). Furthermore, somewhat related articles are:

"Nanocavity-induced trion emission from atomically thin WSe₂", "Emerging photoluminescence from the dark-exciton phonon replica in monolayer WSe₂" and so on. One recent report based on twisted bilayer of WSe₂ shows 4-fold enhancement of up-converted PL (Phonon-assisted up-conversion in twisted two-dimensional semiconductors). Based on these reports, I am still not able to judge what makes this study a novel to be publishable in nature communication. Therefore, I donot recommend this work for nature communication unless authors can demonstrate this concept in some real-time application that highlight the impact of increased up-converted PL from plasmonic nanocavities advantageous over bare monolayer/bilayer WSe₂.

My other comments are below.

1. What is the reason for choosing gold as plasmonic material over other low cost plasmonic materials like copper and silver. What will be the authors comment on choosing silver or copper over gold on up-conversion process? Does author expect similar enhancement?
2. Authors used CW laser as source of excitation, is there any specific reason for that? In Ref 13. authors used femtosecond laser as an excitation source. How the use of these two different sources going to affect the measurements/results interpretation. Apart from the fact that cw laser are safe and donot damage the sample. How due to continue heating effect from cw laser on sample may not cause other nonlinear processes (other than multiphoton absorption) to become an active centre for up-conversion. What is the intensity at the irradiated region?
3. It looks studies were done at room temperature. Room temperature PL emission can be induced by the neutral exciton from direct K-K transition, exciton emission from the conduction band to a lower-lying valence band, as well as negatively charged excitons (ACS Appl. Mater. Interfaces 2020, 12, 9563; Nat. Comm. 2018, 9, 3719). What is author comment on this?
4. While addressing up-conversion process, temperature dependent study plays an important role. Why were temperature dependent measurements not conducted in the present case? What will be the effect of cryogenic temperature on enhanced PL from plasmonic sample?
5. Jadcak et al. (ACS Nano 2021, 15, 19165 – 19174), investigated up-converted PL processes both in high quality WSe₂ monolayer and encapsulated in hBN and a function of excitation energy and in selected temperature. Can authors address, how their assignment of the up-converted PL processes corresponds to the scenario presented by the Jadcak et al. which includes the dark exciton states?
6. First emission peak around 1.65 eV (750 nm) in anti-stoke PL, is approximately 83 meV above the lowest excitation used 1.57 eV (785 nm). How many optical phonons are required to excite electrons from the valence band to the conduction band? How it can be affected by the strong Coulomb effect observed in WSe₂ monolayers.
7. Experiment shows that phonons can exhibit intrinsic chirality in monolayer WSe₂ whose lattice breaks the inversion symmetry and enables inequivalent electronic K and -K valley states. Such phonons are investigated to be chiral through the transient infrared circular dichroism (DOI: 10.1126/science.aar2711). How the nature of these chiral phonons effected by the presences of plasmonic materials.?
8. PL-up conversion peaks shift on varying the excitation energy from 1.5 eV to 1.69 eV as shown in figure 3 a. How authors rule out the possibility that this peak shift in Up-converted PL is not due to anti stoke Raman scattering. Raman processes induce peak shift on varying the excitation energy (Ref. Efficient near-infrared up-conversion photoluminescence in carbon nanotubes).

Reviewer #2:

Remarks to the Author:

The article by NS Müller and coauthors describes the experimental observation of efficient upconversion in WSe₂ by identifying the dark exciton state as a potent excitation path, if coupling to the exciting light is allowed through plasmonics.

In general, I believe this manuscript is suited to be published in Nature Communications. However, I do have a few remarks, of which most are concerning the preciseness of the writing and are in part arising due to the journal not being directed to a narrow community with set lingo. More important points are marked with an "!":

! The authors state in the abstract "that asymmetric nanoparticle shape and precise geometry are key". The used particles for the more efficient NDoM cavities seem to have more symmetries compared to the particles in the NPoM cavities.

Most of the used adverbs are not necessary, and only useful to create emotions or bias the reader. I would recommend removing them. (e.g. "tightly confined plasmonic fields excite the out-of-plane transition dipole", is it the tightness that allows the excitation?; or "even much larger")

! It is stated that decahedral nanoparticles are ideal. How is this claim supported? No data or reference is given for nanoparticles of other (defined) geometry. The given reference evaluates the decahedral particles against what is considered the most ideal spheric particles and against decahedrons of different sizes. To me, it seems, as if the major benefit of the NDoM particles is their uniformity and that their synthesis has been tailored to the optical properties of WSe₂.

Some abbreviations are not introduced, or the introduction could be improved to help the reader remembering (e.g. CAB, EF_{xy}, CTAB solution)

p.2 l.42: A 10⁴ fold enhancement (for a sphere?): is this number relevant to the experiment under consideration, as it was stressed that the "shape and precise geometry" are of importance.

Fig 1b: The image shows a sphere, the caption describes a facet. A precise description (not only in the SI) would be good.

! Fig 1d: Frequency units of 10¹? The caption only describes the histogram. The main text marks the center of the plasmonic mode around 790 nm, while the spectra show peaks from 700-900 nm, with the one of highest frequency at around 800 nm. Clarification is necessary.

p.4 l.5: One or more additional references could be added for the previous observation of the dark exciton in WSe₂ (e.g. from the top of my head from the Toulouse-group: doi.org/10.1038/s41467-021-25747-5) as the observation is not limited by near-field activation.

The methods section seems to have received a bit less tlc compared to the main text.

- The section of the fabrication of the WSe₂ single-layer does not include "WSe₂".
- The gold has been etched. For what reason? And how much of it? (before time and words were taken to describe the 100 nm thickness)
- "the pvp layer is further dissolved". There was no previous dissolution mentioned.
- What is the CAB layer and how thick is it?
- "remove extra surfactant" what does "extra" mean in this sentence?

Finally, I really DO like to see the effort made to collect data to do better than N=2 statistics!

I do recommend publication of the manuscript at hand, after minor revisions.

Reviewer #3:

Remarks to the Author:

The manuscript by Mueller N. S. et al. reports on an anti-Stokes photoluminescence in a monolayer WSe₂ activated by plasmonic nano-cavities through resonant excitation of a dark exciton. Authors show that tightly confined plasmonic fields excite out-of-plane transition dipole of the dark exciton, leading to light emission from the bright exciton at higher energy. This work introduces a new excitation channel for anti-Stokes PL in WSe₂.

Overall the experiments and research in this manuscript contain interests to the material community, and the manuscript is well presented. Therefore I recommend it for publication in the Nature Communication after addressing the following light concerns.

1) The energy splitting between the bright and dark excitons in monolayer WSe₂ have been reported of about 40 meV in Phys. Rev. Lett. 123, 096803, 2019. Could Authors comment on the energy difference of about 50 meV between these peaks in presented results? What is the reason for such a deviation?

2) Have Authors tried to perform comparative PL/upconversion PL experiments under vacuum and ambient conditions? Is there any photo-doping effect which may influence the power dependence of the anti-stokes PL in monolayer WSe₂ (see also Nat Commun. 10, 107 (2019))?

3) Authors write that the largest anti-Stokes PL appears for the energy gain of about 35 meV which matches the A' phonon energy and hints to a phonon-mediated upconversion. As it was suggested in Ref.2, the A' phonon mediates the trion-exciton coupling due to resonant conditions related to the fact that a binding energy of the trion is comparable with the A' phonon energy. Is the same phonon responsible for the coupling of the intra-valley dark and bright excitons? The upconversion process involving the intra-valley dark and bright excitons is not a spin-conserving process. It has been reported that only the E'' phonon couples the intra-valley dark and bright excitons (Nat Commun 11, 618 (2020)). Could Authors comment on it?

Response to Reviewers comments:

We are delighted that the referees emphasise this is ‘an interesting finding’ (reviewer #1), ‘suited to be published in Nature Communications’ (reviewer #2), ‘contain interests to the material community, the manuscript is well presented...’, and ‘recommend it for publication in Nature Comm’ (reviewer #3).

Reviewer #1:

0a) Overall, this could be an interesting finding and may helpful futuristic application in the field of material science and physics. But there are existing reports where such system (i.e. based on plasmonic nanocavity concept) were already addressed where authors demonstrated less than 1000-fold enhancement of up-converted PL due to Purcell effect (author already cited this article in ref 13. Note in this reference some authors are missing too). Furthermore, somewhat related articles are: “Nanocavity-induced trion emission from atomically thin WSe₂”, “Emerging photoluminescence from the dark-exciton phonon replica in monolayer WSe₂” and so on. One recent report based on twisted bilayer of WSe₂ shows 4-fold enhancement of up-converted PL (Phonon-assisted up-conversion in twisted two-dimensional semiconductors). Based on these reports, I am still not able to judge what makes this study a novel to be publishable in nature communication.

> The reviewer helpfully points to several additional relevant publications which we now cite in the revised manuscript (correcting also the cite to [13]). We stress that the key point of our manuscript is to introduce the dark intravalley exciton in WSe₂ as an excitation channel for PL upconversion, which to the best of our knowledge has not been shown before. Plasmonic cavities are thus an enabling tool to couple to the out-of-plane transition dipole of the dark exciton and activate this excitation channel, which introduces a new mechanism for upconversion and goes beyond a simple increase of the local field intensity. Our work therefore clearly differs from Ref. 13 and also other previous publications. Our concept also carries over to other excitons with out-of-plane dipoles, such as localized excitons and excitons in other materials, such as indium selenide, and will therefore have a broader impact.

0b) Therefore, I do not recommend this work for nature communication unless authors can demonstrate this concept in some real-time application that highlight the impact of increased up-converted PL from plasmonic nanocavities advantageous over bare monolayer/bilayer WSe₂.

> To address this, we conducted an entirely new set of experiments which now demonstrates real-time modulation of the upconverted PL – see the new Figure 5 in the revised manuscript (and below) and a new section at the end of the manuscript. Using electrochemical gating in a spectro-electrochemical cell allows us to reversibly modulate the upconverted PL from plasmonic cavities. We observe different modulation of the dark and bright exciton intensities, enabling us to switch their relative strength in-situ. We use this to implement a selective modulation of the upconverted PL, which for example can be used for nanoscale thermal management or super-resolution imaging in future work and broadens the impact of our manuscript.

Figure 5: Selective electrochemical switching of PL upconversion. (a) Sketch of spectro-electrochemical cell (top) and electric double layer at the Au-WSe₂-electrolyte interface (bottom). (b) PL spectra measured with λ_l = 785 nm on NDoM cavity while continuously sweeping the potential from +0.2 V to -0.4 V and back (spectra averaged

for each voltage interval). Modulation on anti-Stokes side (aS) is much larger than on Stokes side (S). (c) Modulation of PL intensity on aS and S sides with $\lambda_l = 785$ nm, and (d) of intensity ratio of dark and bright excitons with $\lambda_l = 633$ nm, while (e) step-wise switching the potential (black) and measured electrical current (blue). Data in (c,d) are recorded on the same NDoM cavity.

1. *What is the reason for choosing gold as plasmonic material over other low cost plasmonic materials like copper and silver. What will be the authors comment on choosing silver or copper over gold on up-conversion process? Does author expect similar enhancement?*

> Indeed this is interesting. We chose gold as it is the most robust plasmonic material against oxidation. Our samples have given data for more than one year without degradation or decrease in enhancement. Our approach however also carries over to other lower cost plasmonic materials, such as silver and copper. We now conducted additional FDTD simulations of the field intensity enhancement under the bottom facet of Au, Ag, and Cu NPoMs (see new Fig. S9 below). A Ag NPoM provides even larger field enhancement ($6 \cdot 10^3$) than Au ($1.5 \cdot 10^3$), while the field enhancement from Cu is slightly lower ($8.5 \cdot 10^2$). The plasmonic resonances of Ag and Cu NPoMs are blue shifted from that of Au NPoMs, but can be tuned into resonance with the WSe₂ excitons by choosing slightly larger nanoparticle sizes. As Ag and Cu are less stable than Au, the metals would have to be encapsulated or the experiments conducted under vacuum conditions.

Figure S9: FDTD simulations of excitation enhancements in NPoM cavities consisting of Au, Ag, and Cu, showing average field intensity enhancements below the bottom facet of the nanoparticle, in a circular area with diameter of 50 nm (see Fig. S8b). All other parameters are the same as in Fig. S8.

2. *Authors used CW laser as source of excitation, is there any specific reason for that? In Ref 13. authors used femtosecond laser as an excitation source. How the use of these two different sources going to affect the measurements/results interpretation. Apart from the fact that cw laser are safe and do not damage the sample. How due to continue heating effect from cw laser on sample may not cause other nonlinear processes (other than multiphoton absorption) to become an active centre for up-conversion. What is the intensity at the irradiated region?*

> This is a useful discussion. We first emphasize that upconversion is not attributed to multiphoton absorption here but to phonon-assisted upconversion which requires a thermal population of phonons. The near-linear power dependence in Fig. S3a,b precludes two-photon absorption. Continued heating from the cw laser is thus desirable here as it increases the thermal phonon population. For our typical cw laser power of 10 μ W, the illumination intensity is 70 μ W/ μ m² in the focused laser spot, which amounts to 300 mW/ μ m² peak intensity in the hotspot of the plasmonic cavity. This is far below the intensity per pulse of 2×10^5 mW/ μ m² used in [13] (without considering also the plasmonic enhancement), which leads to exciton saturation and limits the upconversion enhancement when using pulsed lasers. At lower pulse intensities, our upconversion mechanism still operates fine. Using cw lasers, we avoid the saturation regime and also sample damage, which occurs far more easily with pulsed lasers, as mentioned by the referee. We discuss this now in the methods section.

3. *It looks studies were done at room temperature. Room temperature PL emission can be induced by*

the neutral exciton from direct K-K transition, exciton emission from the conduction band to a lower-lying valence band, as well as negatively charged excitons (ACS Appl. Mater. Interfaces 2020, 12, 9563; Nat. Comm. 2018, 9, 3719). What is author comment on this?

> Indeed, all measurements are conducted at room temperature – see also reply to point 4. We agree that the primary emission at room temperature will occur from neutral A excitons through a direct K-K transition, which we call the “bright” and “dark” excitons in our manuscript (see Fig. 1a). Exciton emission from the conduction band to a lower-lying valence band would correspond to the B exciton, with a transition energy of 2.15 eV ($\lambda = 580$ nm) in WSe₂, which is far away from the emission of the bright A exciton 1.65 eV ($\lambda = 750$ nm) observed here. Upconversion from the A- to the B exciton was indeed previously observed, but only at low temperature (see introduction, Ref. 5). Negatively charged excitons usually only give a weak contribution to the PL signal at room temperature. Ref. 2 reported for example a $1/T$ dependence of the trion absorption rate in the context of PL upconversion. In Nat. Commun. 2018, 9, 3719 all optical spectra were recorded at low temperature ($T=4.2$ K). A previous study reported the activation of trion emission by plasmonic cavities, but this was only observed at temperatures below 160K [Wang et al., Scientific Reports 12, 15861 (2022), Ref. 34 in revised manuscript]. We furthermore do not see signatures of photodoping in the excitation power dependence – see reply to point 2 of Referee 3. We therefore do not believe that charged excitons contribute significantly to the observed signals here. We discuss the role of charged excitons now in the main text on p.4 and in the Supplementary Information on p.4. We also highlight now in the caption of Figure 1a that we refer to A excitons.

4. While addressing up-conversion process, temperature dependent study plays an important role. Why were temperature dependent measurements not conducted in the present case? What will be the effect of cryogenic temperature on enhanced PL from plasmonic sample?

> As temperature is required for the phonon-mediated upconversion observed here, we expect no clear advantage of decreasing the temperature. We would expect a vanishing upconversion signal at cryogenic temperature. This was previously observed in Refs. 2, 13, 17, 35, and 43. Given these many previous reports, and that low-temperature measurements on plasmonic nanoparticle-on-mirror cavities are complex due to the need for high NA collection here, we did not conduct such measurements. However it is important to note, so we added a discussion on p.7 of the main text.

5. Jadczyk et al. (ACS Nano 2021, 15, 19165 – 19174), investigated up-converted PL processes both in high quality WSe2 monolayer and encapsulated in hBN and a function of excitation energy and in selected temperature. Can authors address, how their assignment of the up-converted PL processes corresponds to the scenario presented by the Jadczyk et al. which includes the dark exciton states?

> This clarification is helpful. Jadczyk et al. (Ref. 18) refer to an intervalley electron-electron scattering process which leads to upconversion at twice the energy of conduction band splitting. This process does not require an out-of-plane transition dipole and therefore should not be the dominant process in a plasmonic nanoparticle-on-mirror cavity. Jadczyk et al. refer to a different type of dark exciton with in-plane transition dipole, which is momentum forbidden, i.e. the intervalley dark exciton while we refer to the intravalley dark exciton. In principle, a plasmonic nanostructure can also activate a momentum-forbidden transition but this would require an in-plane field localization on the sub-nm scale which is not realized with our cavities, see Ref. 33 and discussion on p.4. To address this we extended the discussion of momentum-indirect excitons on p.4 of the main text.

6. First emission peak around 1.65 eV (750 nm) in anti-stoke PL, is approximately 83 meV above the lowest excitation used 1.57 eV (785 nm). How many optical phonons are required to excite electrons from the valence band to the conduction band? How it can be affected by the strong Coulomb effect observed in WSe2 monolayers.

> This is an interesting point. The energy difference between excitation at 785 nm and emission at 750 nm is 74 meV, which corresponds approximately to the energy of 2-3 zone-center phonons. Multi-

phonon upconversion was reported previously for TMDs and is very likely the dominant mechanism here. In the experiments with tunable laser excitation we observe the most efficient anti-Stokes PL emission when the excitation laser is detuned from the bright exciton by 20-40 meV, which matches the energies of the optical phonons in WSe₂ (Fig. 3c). The anti-Stokes PL intensity decreases for larger detuning of excitation and emission, as the absorption of multiple phonons is less likely. In this context, the strong Coulomb attraction in TMD monolayers is important, as it enables exciton-phonon coupling, which can scatter bright into dark excitons through the absorption of phonons (see e.g. Refs 40-42). To address this, we extend the discussion of phonon-mediated upconversion on p. 7 of the main text.

7. Experiment shows that phonons can exhibit intrinsic chirality in monolayer WSe₂ whose lattice breaks the inversion symmetry and enables inequivalent electronic K and -K valley states. Such phonons are investigated to be chiral through the transient infrared circular dichroism (DOI: 10.1126/science.aar2711). How the nature of these chiral phonons effected by the presences of plasmonic materials.?

> This is also an interesting point. A possible upconversion mechanism is indeed through exciton-phonon coupling to a chiral E'' phonon – see our reply to point #3 of referee 3 and Ref. 41. As this E'' phonon is optically silent it is unlikely that its nature is affected by the presence of the metal interfaces. The chiral phonon discussed in 10.1126/science.aar2711 is on the other hand infrared active and could in principle be affected by image dipole interactions – see eg. DOI 10.1021/acs.nanolett.2c02806. It is however only excited through a spin-allowed intervalley process involving the bright exciton, and not the spin forbidden intravalley transition studied here. We therefore feel that this discussion is beyond the scope of the current manuscript.

8. PL-up conversion peaks shift on varying the excitation energy from 1.5 eV to 1.69 eV as shown in figure 3 a. How authors rule out the possibility that this peak shift in Up-converted PL is not due to anti stoke Raman scattering. Raman processes induce peak shift on varying the excitation energy (Ref. Efficient near-infrared up-conversion photoluminescence in carbon nanotubes).

> We do not believe that the observed signal originates from anti-Stokes Raman scattering, as the observed peak shift (15 meV) when varying the excitation energy is much smaller than for Raman scattering (160 meV). An anti-Stokes Raman signal would shift in energy exactly as the excitation laser. This is illustrated as a red dotted line superposed on Figure 3a (shown below) which clearly does not match the measured shift (orange line). Electronic Raman scattering of the metal could potentially contribute, which would lead to an exponential tail on the anti-Stokes side of each spectrum. This is however not observed at $\lambda_l = 805$ nm and 730 nm, where the PL emission peak can be distinguished from electronic Raman scattering. Instead, we attribute the small peak shift to the phonon-mediated upconversion mechanism. The upconversion requires coupling to thermally populated phonons, which becomes less likely with increasing energy difference of emission and excitation laser. At long excitation wavelengths (e.g. $\lambda_l = 805$ nm), the anti-Stokes PL is therefore shifted to longer wavelengths. This is also seen in Fig. 1f and Fig. 2b when comparing the Stokes and anti-Stokes emission spectra. We added a discussion on p. 6 of the main text.

The referenced paper does not show upconversion through anti-Stokes Raman scattering – see on page 2 of this reference: “This photon up-conversion does not originate from common coherent two-photon absorption or anti-Stokes Raman processes”.

Fig.3a: with added red dashed line showing expected tuning of aS Raman

Reviewer #2:

1) *The authors state in the abstract "that asymmetric nanoparticle shape and precise geometry are key". The used particles for the more efficient NDoM cavities seem to have more symmetries compared to the particles in the NPoM cavities.*

> Indeed this is helpful to clarify. By 'asymmetric shape' we refer to the broken mirror symmetry of the NDoM cavities along their main axis. NPoM cavities indeed have small imperfections that deviate from a spherical shape, but these are on average far less pronounced than the strong asymmetry of the decahedral nanoparticles. The dark field scattering pattern of NDoM cavities is for example horseshoe-shaped while the scattering of NPoM cavities is a doughnut (Ref. 46). We thus clarify the term to provide additional context (see also our reply to next point below), and reword the abstract.

2) *It is stated that decahedral nanoparticles are ideal. How is this claim supported? No data or reference is given for nanoparticles of other (defined) geometry. The given reference evaluates the decahedral particles against what is considered the most ideal spheric particles and against decahedrons of different sizes. To me, it seems, as if the major benefit of the NDoM particles is their uniformity and that their synthesis has been tailored to the optical properties of WSe₂.*

> Also the reviewer asks for useful details. Electron microscopy images (Fig. 4 and Ref. 44) show that the nanoparticles used for NPoMs contain many nanoparticle shapes, which leads to the large variation in anti-Stokes PL enhancement factors in Fig. 4e, left. As NDoM cavities provide aS PL enhancement factors that are at the top of the NPoM histogram (compare Fig. 4e left and right) we conclude that decahedral nanoparticles are more ideal for this task. This also shows that nanoparticle shape plays an important role in addition to the uniformity and tailored plasmonic resonances. There can be, on the other hand, indeed also other nanoparticle shapes that are suited to achieve large enhancement factors, and we thus now clarify our statement - we reword the introduction on p.2 and add a sentence in the discussion on p.7.

3) *Most of the used adverbs are not necessary, and only useful to create emotions or bias the reader. I would recommend removing them. (e.g. "tightly confined plasmonic fields excite the out-of-plane transition dipole", is it the tightness that allows the excitation?; or "even much larger")*

> As suggested we now reword the abstract as *'The optical near fields of the plasmonic cavities excite the out-of-plane transition dipole ...'*, similar on p1 in the introduction, and on p4. We reword several other sentences in the manuscript, as the reviewer encourages.

4) *Some abbreviations are not introduced, or the introduction could be improved to help the reader remembering (e.g. CAB, EF_xy, CTAB solution)*

> We define all these abbreviations in the revised manuscript and also carefully checked that all other abbreviations are suitably introduced. We presume the reviewer means E_xy, rather than EF_xy.

5) *p.2 l.42: A 10⁴ fold enhancement (for a sphere?): is this number relevant to the experiment under consideration, as it was stressed that the "shape and precise geometry" are of importance.*

Fig 1b: The image shows a sphere, the caption describes a facet. A precise description (not only in the SI) would be good.

> To first answer the second point (which clarifies the first point), Figure 1b shows a sphere that is truncated at the interface to the substrate. This leads to a circular bottom facet of the nanoparticle. In previous work [ref. 32], this was identified as one representative geometry of the NPoM cavities used in our experiments. The optical properties of NPoM cavities can only be properly described when accounting for such bottom facets (see eg. refs. 30 and 32).

The 'near 10⁴-fold enhancement' refers to the field intensity enhancement in the gap between the nanoparticle and the Au mirror (Fig. 1b bottom right panel). The simulated enhancement is not for a sphere, but for a truncated sphere with a circular bottom facet of 32 nm diameter (see dotted lines in Fig. 1b right and Fig. S7). As this is a representative geometry that also reproduces the dark field scattering spectra in FDTD simulations (Fig. S7b), the field enhancement is fully relevant to the experiment. We also now refer to the detailed discussion of the excitation enhancement in the Supplementary Information and the good agreement of simulated and measured enhancement factors (SI Section S6-2). We add details as suggested to the caption of Fig.1b and to the main text.

6) *Fig 1d: Frequency units of 10¹? The caption only describes the histogram. The main text marks the center of the plasmonic mode around 790 nm, while the spectra show peaks from 700-900 nm, with the one of highest frequency at around 800 nm. Clarification is necessary.*

> The reviewer catches a possibly misleading axis label. 'Frequency' refers here to the number of NPoM cavities in each bin of the histogram. We changed the axis label from 'Frequency' to 'Number' in Fig. 1d and 4c. We further clarified the captions of Figure 1 and Figure 4. The sentence *'The dark-field scattering spectra show a broad plasmonic mode with a central wavelength around 790 nm, ...'* on p.3 refers to the histogram bin with the largest number of NPoM cavities, which is between 780 nm and 800 nm (see Figure 1d). The average spectrum for this bin is highlighted with a thicker line in Fig. 1d. To be more general, we rewrote this sentence as suggested.

7) *p.4 l.5: One or more additional references could be added for the previous observation of the dark exciton in WSe₂ (e.g. from the top of my head from the Toulouse-group: doi.org/10.1038/s41467-021-25747-5) as the observation is not limited by near-field activation.*

> This is a helpful suggestion. We cite this now as Ref. 20 and also include additional papers (Refs. 22, 23, 25, 40) which observe the dark exciton. We also rewrite the introduction to highlight other approaches for the activation of the dark exciton than through near fields.

8) *The methods section seems to have received a bit less tlc compared to the main tex.*

> As suggested, we revise the Methods section to address the points below.

- *The section of the fabrication of the WSe₂ single-layer does not include "WSe₂".*

> We add more details about exfoliating WSe₂.

- *The gold has been etched. For what reason? And how much of it? (before time and words were taken to describe the 100 nm thickness)*

> In sample fabrication, the template stripped gold (TS Au) is used twice. In the first step TS Au is for exfoliating monolayers, while in the second step TS Au is used as the substrate for plasmonic cavities. The WSe₂ monolayer is initially exfoliated on a SiO₂/Si substrate with Au tape, using a 100 nm thick Au film as the exfoliation medium. To prepare a clean WSe₂ monolayer, the Au is then etched completely. Afterwards, the WSe₂ monolayer is transferred to the destination substrate which is a TS Au surface.

- *"the pvp layer is further dissolved". There was no previous dissolution mentioned.*

> The word "further" is removed.

- *What is the CAB layer and how thick is it?*

> The CAB is a cellulose acetate butyrate polymer. It is used to transfer WSe₂ from the SiO₂/Si substrate to Au. The thickness is ~100-500 nm, but is completely removed with ethyl ethate after the transfer.

- *"remove extra surfactant" what does "extra" mean in this sentence?*

> We clarify this sentence to *'For SEM characterization, Au nanoparticles were cleaned three times with ultrapure water by centrifugation (3000 rpm) to remove as much of the surfactant as possible.'*

Reviewer #3:

1) *The energy splitting between the bright and dark excitons in monolayer WSe₂ have been reported of about 40 meV in Phys. Rev. Lett. 123, 096803, 2019. Could Authors comment on the energy difference of about 50 meV between these peaks in presented results? What is the reason for such a deviation?*

> This is a good point, referring us to this interesting publication, now cited as Ref 22 in the revised manuscript. As the exciton energies of WSe₂ shift with temperature, we prefer however to compare with previous experiments at room temperature (Refs. 24, 26, 27). The dark-bright splitting in these works varied between 40-60 meV, which is consistent with average splitting $\Delta E_{DB} = 55 \pm 8$ meV observed here (Fig. S5). Variations were previously explained by a large sensitivity of the dark exciton energy to the dielectric environment (Ref. 26). The metal interfaces here lead to image dipole interactions that change the exciton binding energy (Ref. 36), which will be different for the in-plane and out-of-plane transition dipoles of the bright and dark excitons in WSe₂, hence changing the energetic splitting. We discuss this now on p. 4. of the main text.

2) *Have Authors tried to perform comparative PL/upconversion PL experiments under vacuum and ambient conditions? Is there any photo-doping effect which may influence the power dependence of the anti-stokes PL in monolayer WSe₂ (see also Nat Commun. 10, 107 (2019))?*

> Indeed this is also an interesting point. Unfortunately it is problematic to do measurements under vacuum, as our template-stripped Au samples are vacuum incompatible. Jadcak et al. (Nat Comm. 10, 107 (2019), Ref. 35 in revised manuscript) observed a sublinear power dependence of the upconversion intensity in air (slope 0.53) and a linear power dependence in vacuum. This was explained by a depletion of the carrier density under ambient conditions by physisorbed O₂ and H₂O molecules that changes with excitation laser power, and which does not occur in vacuum. We measured a very different excitation power dependence of the upconverted PL, that is slightly superlinear with $P^{1.2}$ both inside and beside the plasmonic cavities. The near linear power dependence hints to very stable conditions, similar to the measurements under vacuum by Jadcak et al. Photodoping is in general possible through hot electron injection from the metal nanocavities – see e.g. Ref. 39, but unlikely here as the nanoparticle and WSe₂ are separated by a layer of ligand molecules. In the case of photo-doping we would expect clear changes of the spectral shape with excitation laser power, i.e. an increase of the low energy peak with laser power because of the formation of charged excitons through photodoping. This was observed by Wang et al. (Sci.Rep. 12, 15861 (2022), Ref. 34 in revised manuscript) at low temperature. We see the opposite trend of an increasing bright exciton intensity with laser power (Fig. S3c, f), which we explain by an increasing thermal population and upconversion. We therefore do not expect a significant contribution of photo-doping in our experiments. We discuss this now in the main text on p.4 of the main text and p. 4 of the Supplementary Information.

3) *Authors write that the largest anti-Stokes PL appears for the energy gain of about 35 meV which matches the A' phonon energy and hints to a phonon-mediated upconversion. As it was suggested in Ref.2, the A' phonon mediates the trion-exciton coupling due to resonant conditions related to the fact that a binding energy of the trion is comparable with the A' phonon energy. Is the same phonon responsible for the coupling of the intra-valley dark and bright excitons? The upconversion process involving the intra-valley dark and bright excitons is not a spin-conserving process. It has been reported that only the E'' phonon couples the intra-valley dark and bright excitons (Nat Commun 11, 618 (2020)). Could Authors comment on it?*

> This is an excellent point. We address it by adding a discussion in the main text of the revised manuscript, where Nat Commun 11, 618 (2020) is cited as Ref. 41, and also another publication Ref. 40 with a more focused discussion on the phonon replica of dark excitons.

Reviewers' Comments:

Reviewer #1:

Remarks to the Author:

Mueller et al. and co-workers reasonably addressed my queries, I am highly impressed from the response letter. I have no more questions except the followings to further improve the quality of the manuscript.

1. I want authors to calculate up-converted PL efficiency and compare it with bare WSe₂ and include that in the manuscript.
2. Furthermore, I would like authors to cite this (<https://doi.org/10.1364/OE.471027>, Unveiling room temperature up-conversion photoluminescence in monolayer WSe₂) recent work on PL up-conversion.
3. I can see in electrochemical switching section only Au-WSe₂ is considered. Can authors compare the results with bare WSe₂?

I would like to see the final draft before recommending it for final publication.

Reviewer #2:

Remarks to the Author:

The authors have addressed all points mentioned by the referees and where they saw need changed the manuscript accordingly.

Generally, I am satisfied with the changes.

One point that I need to stress again is that I would wish for better scientific writing: The text should focus on findings and discuss them neutrally. Explanations should be made to allow the (non-expert) reader to follow, without manipulation. In my opinion, words like "ideal" (the most striking example in the manuscript) used in a general context are not scientific, as - from bare logic alone - one cannot prove this. We can only give evidence that a certain 'property' is not ideal. This fault is clearly seen in the altered discussion section, with the addition of the word "ideal" in line 17 (p.9). Which I feel quite sad about, since the change made on p.7 l.35ff addresses the potential non-idealness but that the given ND-shape is "one of the advantageous geometries", giving the impression that changes are made either from trying to boast the 'relevance' or to satisfy some referee. Neither of which is concerning what would be the correct motivation in terms of science.

I leave the decision to the editor, if they believe whether the manuscript with its wording and claims fits into the scope of the journal.

From a scientific point of view, I can recommend the publication in Nature Communications.

With best regards,
Hans Tornatzky

If the authors wish, I am open to further discuss the topic: tornatzky@pdi-berlin.de

Reviewer #3:

Remarks to the Author:

Authors have considered all my comments and questions. The points raised in the previous round of review have been satisfactorily addressed. Hence, I believe the manuscript can be published in Nature Communications.

Response to Reviewers comments:

We thank all reviewers for assessing our manuscript again and for recommending publication in Nature Communications after minor revisions.

Reviewer #1:

Mueller et al. and co-workers reasonably addressed my queries, I am highly impressed from the response letter. I have no more questions except the followings to further improve the quality of the manuscript.

1. I want authors to calculate up-converted PL efficiency and compare it with bare WSe₂ and include that in the manuscript.

> Based on a comparison to the measured backreflected laser power from Au we calculate the PL efficiency in our experiment. We estimate that with an NPoM cavity 5.10⁻⁴% of the photons that illuminate the sample are upconverted, whereas only 3.10⁻⁶% are upconverted without any NPoM cavity (for the spectra in Fig. 1f). The quantum yield will be much higher, as not all incoming photons are coupled into the nanometer gap of the plasmonic cavity and absorbed by WSe₂, which is however hard to quantify experimentally. The intensity of the enhanced upconverted PL is similar to the intensity of Stokes PL on glass (compare Figs. 1f and S4). For the latter, the quantum yield is known to be on the order of 1% from literature (Ref. 39). For a comparison with the enhanced anti-Stokes PL, uncertainties arise from the unknown ratio of absorption/emission, which can be different because of the different excitation wavelengths (785 nm instead of 633 nm), and the plasmonic cavity. Based on FDTD simulations we expect that the plasmonic cavities mostly enhance the absorption channel, because of the activation of the dark exciton (see Table S1). We therefore estimate a quantum yield of the enhanced upconverted PL on the order of 0.1%. We discuss this now in the revised manuscript and in a new SI Section S5.

2. Furthermore, I would like authors to cite this (<https://doi.org/10.1364/OE.471027>, Unveiling room temperature up-conversion photoluminescence in monolayer WSe₂) recent work on PL up-conversion.

> We appreciate the link to this interesting work, which is now cited as Ref. 14 in the revised manuscript.

3. I can see in electrochemical switching section only Au-WSe₂ is considered. Can authors compare the results with bare WSe₂?

> This is an interesting point. Unfortunately we cannot do such a comparison, as the Au substrate is required for the working electrode in electrochemical gating. We would expect no modulation for WSe₂ on a glass substrate. We however provide reference measurements on bare Au without nanocavities in Figure S12. Electrochemical gating experiments of WSe₂ on transparent ITO substrates were conducted previously, see Ref. 51, but this is however beyond the scope of our work.

Reviewer #2:

The authors have addressed all points mentioned by the referees and where they saw need changed the manuscript accordingly.

Generally, I am satisfied with the changes.

One point that I need to stress again is that I would wish for better scientific writing: The text should focus on findings and discuss them neutrally. Explanations should be made to allow the (non-expert) reader to follow, without manipulation. In my opinion, words like "ideal" (the most striking example in the manuscript) used in a general context are not scientific, as – from bare logic alone – one cannot prove this. We can only give evidence that a certain 'property' is not ideal.

This fault is clearly seen in the altered discussion section, with the addition of the word "ideal" in line

17 (p.9). Which I feel quite sad about, since the change made on p.7 l.35ff addresses the potential non-idealness but that the given ND-shape is “one of the advantageous geometries”, giving the impression that changes are made either from trying to boast the ‘relevance’ or to satisfy some referee. Neither of which is concerning what would be the correct motivation in terms of science.

I leave the decision to the editor, if they believe whether the manuscript with its wording and claims fits into the scope of the journal.

From a scientific point of view, I can recommend the publication in Nature Communications.

> We appreciate this helpful discussion. We agree that the wording that we used in the revised discussion can be improved and thank the referee for pointing this out. We reworded the discussion to avoid the word ‘ideal’. Furthermore, we reworded the manuscript at 12 other places to avoid creating emotions or bias for the reader.

Reviewers' Comments:

Reviewer #1:

Remarks to the Author:

All my queries are very well addressed by Mueller et al. and co-workers. I have no further questions. But still I left final decision on editor if he finds this manuscript suitable to be publishable in Nature Communication or need any further review.

Response to Reviewers comments:

We thank all reviewers for their time and helpful comments which helped us to improve our manuscript, and for recommending publication in Nature Communications.

Reviewer #1:

All my queries are very well addressed by Mueller et al. and co-workers. I have no further questions. But still I left final decision on editor if he finds this manuscript suitable to be publishable in Nature Communication or need any further review.

> Thank you for assessing our manuscript again. We are glad that we could address all your queries. Thank you for your valuable comments that helped us improve our manuscript.